# The Role of PKC-MAPK Signalling Pathways in the Development of Hyperglycemia-Induced Cardiovascular Complications

**DOI:** 10.3390/ijms23158582

**Published:** 2022-08-02

**Authors:** Fatin Farhana Jubaidi, Satirah Zainalabidin, Izatus Shima Taib, Zariyantey Abdul Hamid, Nur Najmi Mohamad Anuar, Juriyati Jalil, Nor Anizah Mohd Nor, Siti Balkis Budin

**Affiliations:** 1Center for Diagnostic, Therapeutic and Investigative Studies, Faculty of Health Sciences, Universiti Kebangsaan Malaysia, Kuala Lumpur 50300, Malaysia; izatusshima@ukm.edu.my (I.S.T.); zyantey@ukm.edu.my (Z.A.H.); ejamdnor@gmail.com (N.A.M.N.); 2Center for Toxicology and Health Risk Research, Faculty of Health Sciences, Universiti Kebangsaan Malaysia, Kuala Lumpur 50300, Malaysia; satirah@ukm.edu.my (S.Z.); nurnajmi@ukm.edu.my (N.N.M.A.); 3Center for Drug and Herbal Development, Faculty of Pharmacy, Universiti Kebangsaan Malaysia, Kuala Lumpur 50300, Malaysia; juriyatijalil@ukm.edu.my; 4Faculty of Health Sciences, University College MAIWP International, Kuala Lumpur 68100, Malaysia

**Keywords:** ERK1/2, JNK, p38, cardiac remodelling, vascular dysfunction

## Abstract

Cardiovascular disease is the most common cause of death among diabetic patients worldwide. Hence, cardiovascular wellbeing in diabetic patients requires utmost importance in disease management. Recent studies have demonstrated that protein kinase C activation plays a vital role in the development of cardiovascular complications via its activation of mitogen-activated protein kinase (MAPK) cascades, also known as PKC-MAPK pathways. In fact, persistent hyperglycaemia in diabetic conditions contribute to preserved PKC activation mediated by excessive production of diacylglycerol (DAG) and oxidative stress. PKC-MAPK pathways are involved in several cellular responses, including enhancing oxidative stress and activating signalling pathways that lead to uncontrolled cardiac and vascular remodelling and their subsequent dysfunction. In this review, we discuss the recent discovery on the role of PKC-MAPK pathways, the mechanisms involved in the development and progression of diabetic cardiovascular complications, and their potential as therapeutic targets for cardiovascular management in diabetic patients.

## 1. Introduction

Diabetes mellitus (DM) has now devoured people on a global scale regardless of socioeconomic status, and it has been estimated that 537 million people worldwide are living with diabetes presently. International Diabetes Federation (IDF) forecasted that this number would rise to 643 million by 2030 and 738 million by 2045 [1]. An uncontrolled state of diabetes could bring about multiple health risks to its sufferers. Over time, diabetes could cause irreversible damage to the heart, nerves, eyes, and kidneys. Often, morbidity among diabetic patients is caused by untreatable complications following chronic or prolonged diabetes.

Past reports have strongly associated cardiovascular diseases (CVDs) with diabetes. Diabetes, a condition brought upon by the failure and/or insufficiency of insulin action that prevents the body from taking up and utilising glucose, is characterised by chronic hyperglycaemia and is a significant factor in the development of CVDs [2,3,4]. This condition provokes many deleterious pathogenic pathways that could increase the risk of morbidity and mortality in a diabetic patient without early intervention.

Many theories prove that chronic hyperglycaemia in diabetes could implicate CVDs, one of them being its direct activation of protein kinase C-mitogen-activated protein kinase (PKC-MAPK) pathways [5,6]. The activation of PKC by hyperglycaemia highlighted PKC as a potential therapeutic target in diabetes mellitus. PKC is a family of isoforms that are subdivided into the classical PKCs (PKCα, PKCβ1, PKCβII, and PKCγ), novel PKCs (PKCδ, PKCε, PKCη, and PKCθ) and atypical PKCs (PKCζ and PKCι) [7]. These isoforms play critical roles in regulating cell proliferation, apoptosis, differentiation, angiogenesis and tumour development. The stimulation of PKC activates multiple pathways, including the mitogen-activated protein kinase (MAPK) via activation of MAPK kinases (MAPKKs) [8]. Overt activation of PKC-MAPK has been widely debated to play a significant role in the development of many diabetes-induced pathologies, including nephropathy and diabetic liver [9]. However, the role of PKC-MAPK activation in the development of diabetes-induced cardiovascular complications has yet to be discussed in detail. Therefore, in this review, we aimed to elucidate the role of the overt activation of PKC-MAPK signalling pathways in the development of diabetes-induced cardiovascular complications.

The literature search was completed using the PubMed database and relevant keywords (protein kinase C; mitogen-activated protein kinase; protein kinase C- mitogen-activated protein kinase; p38; ERK1/2; JNK; cardiovascular; diabetes mellitus; vascular; heart; hypertrophy; remodelling; fibrosis; inflammation; oxidative stress) were used as a search strategy. This article will highlight the role of PKC-MAPK signalling pathways in the development of diabetes-induced cardiovascular complication and its potential as a therapeutic target for cardiovascular management.

## 2. Protein Kinase C-Mitogen Activated Protein Kinase (PKC-MAPK) Signalling Pathways in Diabetes

Studies have demonstrated that essential factors of hyperglycaemia promote cardiovascular complications, which include the promotion of oxidative stress, inflammation, and apoptosis on myocardial and vascular tissues, eventually leading to irreversible functional and structural damage upon persistent stimulation. These three pathophysiological mechanisms result from the hyperglycaemic activation of various signalling pathways, including polyol and hexosamine pathways, and notably in this review, protein kinase C [10]. Hyperglycaemic activation of PKC signalling leads to increased endothelial inflammation, endothelial relaxation dysfunction, and foam cell formation, which induces the expression of permeability-enhancing factor vascular endothelial growth factor (VEGF) and decreases nitric oxide (NO) generation in vascular smooth muscle cells [11,12]. All the results emphasise the unique characteristics of diabetes-associated atherosclerosis. In fact, it is also well established that increased myocardial PKC activity could be observed after chronic or persistent diabetes [13,14].

### 2.1. PKC Subfamilies

PKC is a group of serine/threonine protein kinases that play many roles in various cellular activities across different systems. Their activation regulates cell growth, differentiation, and cell death as well as transformation [15,16,17]. Each PKC isoform mediates various biological and cellular activities. Structurally, PKC contains an NH2-terminal (regulatory domain) and COOH-terminal (catalytic domain). All PKC have a pseudosubstrate side that keeps them in their inactive forms (Figure 1). What differentiates each PKC from another is its structure and cofactor requirement as well as substrate specificity. According to these variations, PKCs are divided into three major groups (i) the classical PKC (cPKCs) including the α, βI, βII, and γ isoforms; (ii) the novel PKC (nPKCs) including the θ, η, ε, and δ isoforms; and lastly (iii) atypical PKC (aPKCs) including ς and ι\λ isoforms.

The members of the cPKC subfamily have conserved (C1–C4) and variable (V1–V5) regions, and they require calcium, phosphatidylserine, and diacylglycerol (DAG) or phorbol ester to be activated. nPKC differs from cPKC in that they lack a C2 homologous domain and are not activated by calcium. PKC is kept in an inactive form in the absence of its cofactors by the binding of the PS to the substrate binding cavity. aPKC is insensitive to DAG, phorbol esters, and calcium as they lack both C2 and half of the C1 homologous domain. Therefore, in diabetic conditions with high blood glucose levels, cPKC and nPKC are majorly activated. Unlike other kinases such as Akt, activation of PKCs does not require acute phosphorylation of the enzyme: phosphorylation necessary for catalytic competence occur shortly after synthesis, and the enzyme is constitutively phosphorylated at these sites [18,19]. As a result, changes in phosphorylation do not provide an indication of PKC activity; rather, signalling-induced translocation of the enzyme to the membrane/particulate fraction represents the most reliable means of monitoring the kinase activation.

It is well known that activated PKC triggers the MAPK cascade. Recently, the importance of MAPK in diabetic complications has been discussed. Some recent reports studied the role of PKCs and MAPKs in the development and progression of diabetic nephropathy. PKC was reported to directly activate MEK and ERK, leading to overexpression of TGF-β that promotes cardiomyocyte remodelling, especially hypertrophy, which only worsens over time. Activation of the PKC-MAPK pathway has been well established by studies at the protein level, that is, by detecting increased serine/threonine phosphorylation in high glucose conditions.

### 2.2. Activation of PKC in Hyperglycemic Condition

The activation of cPKCs and nPKCs isoenzymes requires correct phosphorylation and the presence of cofactors such as diacylglycerol (DAG) and Ca^2+^. Upon proper phosphorylation, increment in both Ca^2+^ and DAG will induce its translocation from the cytosol onto the membrane to exert biological actions [20,21]. In diabetic conditions, activation of PKCs mainly occurs due to the high production of DAG and an increase in oxidative stress (Figure 2).

The abundance of DAG produced in hyperglycaemic conditions could only further promote the excessive activation of PKC as it is one of the cofactors needed to activate both cPKCs and nPKCs [22]. In diabetic conditions, an increase of DAG contents occurs through a few pathways. One of the most common ways is by increasing de novo DAG synthesis from the glycolytic intermediate dihydroxyacetone phosphate (DHAP) through its reduction to glycerol-3-phosphate (G3P) and stepwise acylation [23]. Increased synthesis of DAG is caused by inhibition of the glycolytic enzyme glyceraldehyde-3-phosphate dehydrogenase (GAPDH) from the conversion of glyceraldehyde 3-phosphate (GAP) into the glycolysis-energy production pathway [24]. Thus, the increment of GAP metabolites results in increased flux of DHAP into DAG production. In addition, DAG can also be derived from the hydrolysis of phosphatidylinositides from the metabolism of phosphatidylcholine by phospholipase C (PLC) [25].

Hyperglycaemia-induced increases in oxidants such as H_2_O_2_ are also known to activate PKC. In fact, the presence of antioxidants prevented PKC activation, as reported by Tuttle and colleagues [26]. Complete inhibition of PKC activation by several distinct antioxidants implicates oxidative stress as one of the primary PKC activators as seen in mesangial cells [26]. Although oxidative stress does not induce the formation of DAG, Konishi and colleagues [27] found that H_2_O_2_ induces PKC modification by tyrosine phosphorylation, thus activating the enzyme. This mainly applied to PKCα and PKCδ [27].

### 2.3. Roles of PKC in Diabetic Cardiovascular Pathology

Understanding the distribution of PKC isoenzymes in cardiovascular function requires knowledge of the expression of PKCs in the cardiovascular tissues. PKCs, after all, are the central enzymes in the regulation of cell growth and hypertrophy aside from playing a major role in signal transduction in the heart. They also regulate endothelial function. It is a consensus that activation of PKCs may exert different effects in both normal and pathological conditions.

Among the different classical PKC isoenzymes expressed in cardiac tissues, PKCα is the most predominant member and has been widely discussed in literature. While it has been shown to play a role in regulating cardiac muscle contractility [28], PKCα has also been inferred to be over-expressed in cardiac pathology and transition to heart failure [29]. Both PKCβ and PKC𝛄 also were shown to play minor roles in heart physiology; similarly, they were upregulated in dysfunctional heart and endothelial [30,31,32,33,34].

In contrast, novel PKC isoenzymes were critical in protecting the heart against ischemia and reperfusion. Deletion of the gene encoding for PKCθ was followed by a reduction in fractional shortening, increased fibrosis, and elevation of cardiac injury markers [35]. However, in chronic pathology such as diabetes, in which overexpression of the kinase is persistent, cardiac hypertrophy was observed [36] and is believed to be a result of its activating action on JNK and p38 MAPK pathways [35]. Takeishi et al. [36] suggested that the preserved systolic function of the hypertrophic heart could result from PKCθ activation. However, PKCδ was reported to show contradicting effects. While it was found to be cardioprotective against reperfusion injury [37], PKCδ also was found to mediate cardiomyocyte apoptosis by promoting mitochondrial ROS production in the hyperglycaemic heart [38].

## 3. Mitogen-Activated Protein Kinases (MAPKs)

The MAPK kinases is a family of serine/threonine kinases that play roles in signal transduction pathways that regulate intracellular events, such as acute responses to hormones and developmental changes in organisms [39]. Complex regulatory mechanisms are utilised to direct the functional outcomes mediated by MAPKs. MAPK pathways can be activated by several stimulators, including extracellular stressors, cytokines, growth factors, and PKCs among others [40,41,42]. Following stimulation, activation of MAPK requires a three-tiered kinase cascade; in which a MAPK kinase kinase (MAPKKK, MEKK, MAP3K or MKKK) activates a MAPK kinase (MAPKK, MEK, MAP2K or MKK), which in turn activates the targeted MAPK through serial phosphorylation (Figure 3). Once activated, the MAPK phosphorylates diverse substrates in the cytosol and nucleus to execute intended biological responses [43,44].

MAPKs are involved in a wide range of biological processes, including cell proliferation, differentiation, metabolism, motility, survival, and apoptosis [39]. Hence, dysregulation or improper functioning of these cascades is involved in the induction and progression of diseases. It is one of the signalling pathways involved in the development and progression of cardiac and vascular complications. It is activated in response to a wide variety of extracellular stimuli and induces changes in critical intracellular processes promoting cell growth, apoptosis, and transformation. These extracellular stimuli include cellular stress, adhesion molecules, and neurohormones as well as PKC [45,46,47]. Increasing evidence shows that different members of the MAPK family are critically involved in regulating signalling pathways, ultimately leading to cardiac remodelling and vascular complications [48,49,50,51,52,53]. These biological events result from signal transduction and regulation by the four MAPK subfamilies, which are extracellular-signal-regulated kinases (ERK1/2), c-Jun NH2-terminal kinases (JNK 1, 2 and 3), and p38 MAPK.

It is a well-established fact that activated PKC triggers the MAPK cascade. Recently, the importance of MAPK in diabetic complications has been widely studied. From some recent reports that examined the role of PKCs and MAPKs in the development and/or progression of diabetic nephropathy, PKC was reported to directly activate ERK and JNK pathways [54,55], leading to overexpression of TGF-β that promotes cardiomyocyte remodelling, especially hypertrophy, which only worsens over time [56,57].

### 3.1. Extracellular-Regulated Kinase (ERK) 1/2

ERK1 and ERK2 are 83% identical, share most signalling activities, and, as a result, are usually referred to as ERK1/2 [58,59]. ERK1/2 regulates many cell processes, including proliferation, transcription, differentiation, migration, and cell adhesion [60]. Due to its wide range of roles in biological processes, dysregulation of ERK1/2 has been shown to significantly take part in the development of various pathologies, including diabetes and cardiovascular diseases [61,62,63].

ERK1/2 is activated via a three-tiered kinase cascade by both extracellular and intracellular stimuli, and it could also be triggered by G-protein-coupled receptors [64], cytokines [65], microtubule disorganisation [66] and other stimuli including PKC [67]. Cheng et al. [68] reported that PKC-α and PKC-ɛ act as Raf-1 activators, leading to a prolonged effect on the ERK1/2 signalling pathway. Activated Ras recruits and activates Raf (MAP3K) at the plasma membrane. Once activated, Raf phosphorylates and activates MEK1/2 (MAP2K). MEK1/2, in turn, activates ERK1/2 by phosphorylation at the Thr and Tyr residues in the conserved Thr-Glu-Tyr motif within its regulatory loop. Activated ERK1/2 can then phosphorylate downstream proteins in the cytoplasm or nucleus, including many transcription factors.

### 3.2. c-Jun N-Terminal Kinase (JNK)

JNK plays a role in several biological processes, including cell proliferation, differentiation, apoptosis, cell survival, actin reorganisation, cell migration, metabolic programming, and inflammation [69,70,71,72,73]. This translates into JNK’s physiological role in the development of various pathologies upon its aberrant signalling. Activation of the JNK pathway occurs in response to several different stimuli. As a stress-activated protein kinase, JNK responds most robustly to inflammatory cytokines and cellular stresses such as heat shock, hyperosmolarity, ischemia-reperfusion, UV radiation, oxidant stress, DNA damage, and ER stress [74]. After the cell is stimulated, signalling occurs, eventually leading to the activation of the first tier. The MAP3Ks that can activate JNKs are MEKK1, MEKK2, and MEKK3, as well as mixed lineage kinase 2 and 3 (MLK2 and MLK3) among others. These kinases then activate the subsequent MAP2Ks in the cascade, which are MKK4 and MKK7. MKK4/7 then activates JNK by phosphorylation.

Given the role of MKK4 and/or MKK7 in activating JNK, it is essential to define the relationship between these kinases and PKC in JNK activation. Inhibition of MKK4 and/or MKK7 expression attenuated PKC’s ability to activate JNK, suggesting that MKK4 and MKK7 are required for PKC-dependent JNK activation [75].

### 3.3. p38

Like the former MAPK subfamilies mentioned earlier, p38 kinases also play numerous biological roles in normal physiology and pathology. Most prominently, p38 signalling is involved in the immune response, promoting the expression of proinflammatory cytokines, cell adhesion molecules, and other inflammatory-related molecules and regulating the proliferation, differentiation, and function of immune cells [76,77]. p38 also plays a role in many biological processes in the cardiovascular system, namely, apoptosis, cell survival, cell cycle regulation, differentiation, senescence, and cell growth and migration [78]. Physiologically, this implies that p38 plays a role in chronic inflammatory diseases, including cardiovascular disease.

p38, as one of the stress-activated kinase, responds to many of the same stimuli as JNK, as well as some that are unique to p38. Stress stimuli such as UV radiation, heat, osmotic shock, pathogens, inflammatory cytokines, growth factors, and others can activate p38. p38 plays a complex and involved role, responding to over 60 different extracellular stimuli in a cell-specific manner, making elucidating its exact functional role in vivo difficult. PKC has been shown to directly activate p38, which also functions upstream of p38 [79,80,81]. The canonical pathway of p38 activation is the same as that of ERK and JNK, regardless of the exact stimuli. Several upstream kinases are implicated in the phosphorylation cascades leading to the activation of p38, including MEKK1–4, TAK1, and ASK1 at the MAP3K level and MKK3, −6, and, possibly, −4 at the MAP2K level are implicated in the phosphorylation cascades leading to p38 activation. These MAP2Ks work by phosphorylating p38’s conserved Thr-Gly-Tyr motif [44].

### 3.4. Activation of MAPK by PKC

Studies have consistently demonstrated that aberrant regulation of MAPK cascades and its upstream activators predispose to the development of numerous pathological conditions. The mechanisms involved in the activation events for the MAPK cascade have been studied extensively. As discussed, all MAPK subfamilies are activated upon stimulation of various stimuli. Regardless, all stimuli are to be able to phosphorylate within the activation loop of the kinase to achieve activation, and the immediate upstream kinases catalyse these. In that regard, evidence showed that PKC isoforms could activate MAPK pathways via multiple mechanisms, as overexpression of both PKCδ and PKCη has previously been shown to activate MAPKs [82,83].

Among the most commonly discussed activation mechanisms of PKC on MAPK would be via Ras/Raf activation [84], especially on the PKC/ERK pathway [85]. Activated Ras by PKC prompts recruitment of Raf onto the membrane, thereby activating it. Activated Raf then acts on MEK dual specificity kinase that phosphorylates ERK on both the Thr and Tyr residues [86,87]. In addition, activated Raf activates other MAP3K kinases that would subsequently activate MAP2Ks and MKK/MEK, which then phosphorylate the MAPK effectors JNK and p38 [88,89,90]. In addition, Receptor for Activated C Kinase 1 (RACK1) was also found to be an essential regulator of JNK signalling, as it converges both MKK4 and/or MKK7 and PKC components [75].

## 4. Effects of Activation of PKC-MAPK Pathways to Cardiovascular in Hyperglycaemic Conditions

It is already a well-established premise that prolonged exposure to hyperglycaemia in diabetic conditions leads to the development and progression of cardiovascular complications that only worsen with time if early intervention is not initiated. Cardiovascular complications mainly result from the unrestrained remodelling of both cardiac and vascular structures that prompt functional disturbances in both systems. As such, hyperglycaemia induces cardiac and vascular structural changes by promoting oxidative stress, inflammation, and apoptosis to the cardiac and vascular components. Hyperglycaemia-induced pathways, including PKC-MAPK pathways, stimulate all these pathological mechanisms. The MAPK signalling pathway is related to inflammatory, oxidative, and apoptotic processes and therefore has a crucial role in the development of diabetes-induced cardiovascular complications. Figure 4 summarises the mechanism involved in the development of diabetic cardiovascular complications via the PKC-MAPK pathway.

### 4.1. Cardiovascular Oxidative Stress

Oxidative stress is one of the major aetiologies for various complications in diabetes, including both cardiac and vascular complications. The redox imbalance in the cell causes oxidative stress, in which the endogenous antioxidant capability is overwhelmed by the increasing ROS generation [91]. The accumulation of reactive oxygen species (ROS) and their persistent high levels is enhanced by impaired antioxidant defence and leads to diabetes-associated cell inflammatory and apoptotic responses. Activated PKC would lead to the generation of reactive oxygen species (ROS) via activation of nicotinamide adenine dinucleotide phosphate (NADPH) oxidase [92,93]. This is proven by the fact that inhibitory action on PKC-β was found to reduce the induction of several subunits of NADPH oxidase in a high glucose environment [94,95].

Superoxides produced by activated NADPH oxidase could readily react with nitric oxide (NO), forming highly reactive and damaging peroxynitrite species [96]. Cellular resources of superoxides within the heart include cardiac myocytes, endothelial cells, and neutrophils. The reactive, unstable superoxide could lead to cellular damage via several mechanisms (oxidation, interference with NO, and modulation of detrimental intracellular signalling pathways). Direct damage exerted by these superoxides causes cardiac dysfunction by inducing apoptotic and inflammatory responses, thereby stimulating the cardiac remodelling process [97]. The cardiac remodelling process is a compensatory response against structural and functional damage that include activation of matrix metalloproteinase (MMP) to alter the architecture of the extracellular matrix and modulation of signal transduction pathways that initiate cardiomyocyte hypertrophy in order to preserve normal functions of the heart [98,99]. However, uncontrolled and excessive cardiac remodelling in persistent diabetic conditions further disturbs the ability of the heart to contract and relax efficiently, resulting in the development of diastolic and systolic dysfunctions along with the appearance of prominent cardiac structural changes [100].

In the vessels, the imbalance between NO bioavailability and the accumulation of ROS in hyperglycaemic conditions induce alterations in the vascular function, eventually leading to endothelial dysfunction [101]. Indeed, hyperglycaemia-induced generation of superoxide anion inactivates NO to form peroxynitrite, a powerful oxidant which quickly penetrates across phospholipid membranes and induces substrate nitration. Protein nitrosylation blunts the activity of antioxidant enzymes and endothelial NO synthase (eNOS) [97].

### 4.2. Cardiovascular Inflammation

A maladaptive proinflammatory reaction has been implicated in the development of diabetic cardiovascular complications. Activation and expression of proinflammatory cytokines, such as tumour necrosis factor α, interleukins 6 and 8, monocyte chemotactic protein 1, adhesion molecule intercellular adhesion molecule 1, and vascular adhesion molecule 1, all contribute to cardiac oxidative stress, remodelling and fibrosis, and diastolic dysfunction [102]. The nuclear transcription factor, NF-κB, regulates cytokine expression. Finally, toll-like receptor-4 also plays a crucial role in initiating NF-κB, proinflammatory, and innate immune system responses. These proinflammatory reactions occur in numerous populations of cardiac cells, coronary endothelium and smooth muscle cells, as well as fibroblasts and cardiomyocytes [103,104]. Persistent hyperglycaemia was found to induce the expression of these cytokines in the heart by activating the MAPK pathways (JNK and p38-MAPK) [105,106,107], causing cardiac damage and exacerbating leukocyte infiltration and accumulation at the affected site [108].

Moreover, oxidative stress is further exacerbated by persistent hyperglycaemia and hyperlipidaemia, in which PKC activation also plays a role by activating mitochondrial NADPH oxidase [109]. Inhibition of Nrf-2 by ERK 1/2 exaggerates oxidative stress [110], activating NF-κB [111,112]. This evidence shows that via activation of NF-κB, which promotes the production of pro-inflammatory cytokines and causes hyperglycaemia and hyperlipidaemia, the PKC/ERK pathway can induce cardiac and vascular inflammation.

### 4.3. Cardiovascular Apoptosis

In diabetes, several mechanisms, including hyperglycaemia, hyperlipidaemia, oxidative stress, increased circulating inflammatory cytokines, mitochondrial dysfunction, caspase activation, altered expression of anti- and pro-apoptotic proteins, and ER stress can increase the rates of apoptosis in cardiomyocytes [113]. Hyperglycaemia-induced cardiomyocyte apoptosis occurs through activation of the mitochondrial cytochrome c-stimulated caspase-3 pathway, which may be triggered by high glucose-derived ROS production [114]. In addition to the direct effect of oxidative stress on the activation of mitochondrial cytochrome c stimulated caspase-3 pathway, DNA damage induced by ROS can also activate p53-dependent apoptosis [115,116]. The p53 protein was shown to induce apoptosis by activating pro-apoptotic members, including Bax, and interacting with and inhibiting anti-apoptotic members of the Bcl-2 family, thereby stimulating the mitochondrial pathway of apoptosis [117].

In the mitochondrial death pathway, the pro-apoptotic protein Bax is induced and inserted into the mitochondrial membrane to form a channel allowing the release of cyt c and subsequent activation of caspase-3 through the formation of the cyt c/APAF-1/caspase-9-containing apoptosome complex [118]. The effector caspase-3 proteolytically cleaves cellular proteins and DNA, causing the apoptotic demise of the cell [119,120]. Activated caspase-3 can lead to DNA fragmentation by activating the specific nuclease caspase-3-activated DNase (CAD), allowing CAD to enter the nucleus, where its activity leads to DNA fragmentation [121]. Consequently, the membrane phospholipid phosphatidylserine, a phospholipid embedded in the plasma membrane oriented toward the cytosol, inverts and becomes exposed on the cell’s surface. The phosphatidylserine, which acts as a “eat me’’ signal, attracts the phagocytic cells that, in turn, engulf the apoptotic cells [120]. Additionally, caspase- 3-mediated PKCδ cleavage is mandatory for oxidative stress-induced vascular smooth apoptosis, and PKCδ acts both upstream and downstream of caspase-3 [122].

## 5. Involvement of PKC-MAPK Pathway in Diabetic Heart Complications

### 5.1. PKC-MAPK Pathway in Cardiac Hypertrophy

Cardiac complications mainly arise following alterations to its structure and functions, and in the context of diabetes, develop following uncontrollable persistent hyperglycaemia. Cardiac remodelling is a compensative process following alteration of the structure of the heart in response to mechanical, electrical, or neurohormonal stimuli, and diabetes is one of the triggering factors [123]. Both cardiac hypertrophy and fibrosis refer to cardiac remodelling processes. Cardiac hypertrophy is a thickening of the interventricular wall and/or septum in the cells, and it involves multiple complex and progressive alterations of the heart’s geometry [124]. Although short-term subcellular changes associated with cardiac hypertrophy may be beneficial, the cardiac system becomes maladaptive when sustained for longer intervals. This eventually leads to decompensation resulting in fibrosis, apoptosis, and cardiac remodelling, among other cardiac diseases, before transitioning to heart failure. Hypertrophy is therefore an early indication during the clinical course of heart failure and plays a significant risk factor for subsequent cardiac death.

Numerous studies have suggested that the Ras/Raf/MEK1/ERK signalling pathway contributes to the development of cardiac hypertrophy. Hunter et al. [125] first demonstrated that transgenic expression of a constitutively active Ras in the mouse heart resulted in left ventricular hypertrophy accompanied by cardiomyocyte hypertrophy but not an increase in cardiac fibrosis. Similar overactivation of Ras has been observed upon overexpression of MEK 1, the upstream activator of ERK1/2. In vitro, constitutively active MEK1 promotes cardiomyocyte hypertrophy, whereas dominant negative MEK1 inhibits this response [126]. In vivo, activated cardiac-specific MEK1 expression promotes hypertrophy [127]. However, unlike Ras overactivation, the MEK1 transgenic heart exhibits no increase in fibrosis and preserved cardiac function, indicating that MEK-ERK may not be the most crucial downstream signalling pathway for Ras-induced pathological remodelling.

In addition to these gain-of-function strategies, Mutlak et al. [128] demonstrated that inhibition of the ERK pathway by dominant negative Raf reduced hypertrophy and foetal gene induction in response to pressure overload. In addition, Yamaguchi et al. [129] also demonstrated that cardiac-specific deletion of c-raf-1 causes heart failure in the absence of hypertrophy. Despite the apparent lack of hypertrophy, a significant increase in apoptosis was observed in response to Raf inactivation. This is consistent with the observation that overactivation of the ERK pathway induces hypertrophy and partial apoptosis resistance [130].

Several reports have associated PKC activation with either cardiac hypertrophy, heart failure, ischemic injury, or agonist stimulation. Overexpression of either wild-type or a constitutively active deletion mutant of PKCβ in the experimental heart was reported to induce cardiomyopathy [131,132]. Many studies implicated the role of ERK1/2 activation in the detrimental processes of oxidative stress, inflammation, remodelling, and apoptosis of the heart. Cardiomyocyte apoptosis is one of the crucial components in early cardiac responses that may lead to the devastating complications of cardiomyopathy. Interestingly, studies showed that ERK1/2 activation might exert both pro and anti-apoptotic effects on the heart. This is due to the role of ERK1/2 in regulating cell proliferation and differentiation. Thus, its upregulation may protect the heart against acute cardiac injuries, such as ischemic reperfusion and myocardial infarction [133]. This is supported by the protective effect of ERK1/2 against mitochondrial fragmentation and cardiomyocyte apoptosis in an infarcted heart. This protective effect of ERK1/2 shows that ERK1/2 does regulate hypertrophic response. However, this response depends heavily on the duration and strength of the signals and crosstalk with other intracellular signals [134]. That explains why, in hyperglycaemic conditions whereby DAG-PKC is extendedly prolonging ERK1/2 activation, its hypertrophic, pro-apoptotic, and ROS-productive traits hasten the development of cardiomyopathy, especially hypertrophy.

In addition, Wang et al. [135] reported that overactivation of JNK by MKK7, an upstream MAP2K, leads to a hypertrophic characteristics in cultured cardiomyocytes. Similarly, another upstream MAP2K, MKK4, suppressed the endothelin-1-induced hypertrophic response in cultured myocytes [136]. Similarly, early in vivo studies in rats revealed that dominant-negative MKK4 inhibited JNK activity as well as pressure overload-induced hypertrophy [137]. These findings suggest that JNK activity is involved in the promotion of cell hypertrophy. An in vivo study found that targeting p38 in the heart did not result in significant cardiac hypertrophy. Transgenic overexpression of MKK3 or MKK6 in the heart, on the other hand, increased interstitial fibrosis, ventricular wall thinning, and premature death due to cardiac failure. These findings are supported by Klein et al. [138], in which loss of PKC-ε resulted in increased activation of p38 in the myocardium following pressure overload. These animals displayed a similar phenotype to the MKK3/MKK6 animals, including no increase in hypertrophy but a significant increase in fibrosis and impaired diastolic function. These findings indicate that, in vivo, p38 activity alone is not sufficient to promote cardiomyocyte hypertrophy. Conversely, initial studies involving cardiac-specific p38 dominant negative transgenic mice showed that loss of p38 activity either did not affect hypertrophy [139] or sensitised the heart to hypertrophy in response to pressure overload. Cardiac-specific deletion of p38 did not alter pressure overload hypertrophy. It resulted in a similar degree of myocyte hypertrophy between p38 CKO and wild-type animals following pressure overload [140]. However, these mice exhibited an increase in apoptosis, fibrosis, chamber dilation, and reduced LV function. Taken together, these findings would indicate that p38 activity is not involved in promoting hypertrophy in vivo but may play an important role in pathological remodelling.

### 5.2. PKC-MAPK Pathway in Cardiac Fibrosis

Activation of the JNK pathway in the heart resulted in lethal restrictive cardiomyopathy with selective extracellular matrix remodelling. Prolonged activation of JNK activity in the heart was also associated with abnormal gap junction structure, loss of the main component (connexin-43), and slowed conduction velocity in the heart [141]. Recent evidence suggests that the loss of gap junctions in JNK- activated hearts is associated with the loss of connexin-43 protein expression as well as improper intracellular targeting [142]. On the other hand, deletion of JNK1 in the heart resulted in an increase in fibrosis following pressure overload [143]. Similarly, chronic treatment with a JNK inhibitor increased apoptosis and cardiac fibrosis in the cardiomyopathic hamster model [144].

JNK has also been implicated in promoting cardiac remodelling downstream of various pathways. For example, ASK-1/JNK has been shown to play a role in β-adrenergic-induced cardiac remodelling and apoptosis in vivo [145]. Hsp20, a protein with known cardioprotective effects [146], inhibited the activation of JNK in this setting. In a rat model of pressure overload, cardiac remodelling was inhibited by inhibiting MAPK signalling, including JNK activity, by upregulating mitogen-activated protein kinase phosphatase [147]. Finally, MMPs are well known to contribute to cardiac remodelling. Recent in vitro studies have shown that in response to β-adrenergic signalling, extracellular matrix metalloproteinase inducer (EMMPRIN) expression and MMP-2 activity were increased in a JNK-dependent manner in cardiomyocytes [148]. These findings are supported by other in vitro work in which JNK activation in H9c2 cardiomyoblasts resulted in the up-regulation of MMP-2 (but not MMP-9) activity [149]. Likewise, in vivo studies on the loss of β1-integrins showed that increased JNK activity was associated with increased MMP-2 but not MMP-9 activity, which corresponded with less cardiac fibrosis in this setting. ROS signalling is emerging as an important player in cardiac remodelling. Since JNK activation is a downstream consequence of ROS induction, there might be a more significant role for JNK in cardiac remodelling than originally thought.

As with JNK and ERK1/2, the other stress-activated MAPK, p38, appears to play an important role in cardiac remodelling. Liao et al. [150] discovered that targeted p38 activation in the myocardium led to restrictive cardiomyopathy with significant interstitial fibrosis. In this setting, p38 induced cytokine release from myocytes, including TNF-α and IL-6 [151]. Interestingly, in these in vitro studies, blocking p38 activity did not appear to prevent cytokine production in the myocytes, only their release from the cell. Proinflammatory cytokines such as TNF-α and IL-6 are known to act in both autocrine and paracrine fashions. Acting in an autocrine manner, they are known to have negative inotropic effects [151]. Acting in a paracrine fashion, they play a significant role in myocardial remodelling [152]. These initial studies are supported by the findings of Tenhunen et al. [153], in which DNA microarray analysis of animals with cardiac-specific overexpression of p38 revealed that genes related to inflammation and fibrosis were among the most significantly upregulated.

P38 is also activated by proinflammatory cytokines, including transforming growth factor (TGF)-β. This type of p38 activation also contributes to cardiac remodelling. Indeed, p38 is activated via a TGF-β1/TAK1-dependent mechanism in myocytes following myocardial infarction in rats [154]. As discussed previously, p38 activity can induce cytokine production, thus creating a type of feed-forward mechanism for cytokine action and production. This autocrine and paracrine signalling can lead to the recruitment and proliferation of cardiac fibroblasts and inflammatory cells, resulting in remodelling. Recent work in aged hypertensive rats (which naturally develop significant amounts of cardiac fibrosis) has shown that treatment with a TGF-β antagonist dramatically reduces both hypertrophy and interstitial fibrosis [155]. A decrease in phosphor-p38 levels accompanied this. This fits with other studies showing that inhibition of p38 reduces remodelling following myocardial infarction [156].

### 5.3. PKC-MAPK Pathway in Endothelial Dysfunction

Endothelial dysfunction is one of the initiating factors in the development of macro- and microvascular complications in diabetic conditions due to endothelial cells’ role in regulating vascular homeostasis [157]. Endothelial dysfunction in macrovascular complications is characterised by reduced nitric oxide bioavailability and increased production or action of endothelium-derived vasoconstrictor [158]. On the other hand, in microvascular complications, endothelial dysfunction is portrayed by a reduced release of nitric oxide, excessive oxidative stress, increased production of inflammatory factors, abnormal angiogenesis, and impaired endothelial repairs [159]. Huang et al. [160] demonstrated that endothelial dysfunction in diabetic mice was mediated by both p38 and JNK activation with significant impairment in endothelial nitric oxide synthase (eNOS). In fact, phosphorylated vascular p38 and JNK were found to directly enhance the production of superoxides, which scavenge NO and contribute to its reduced bioavailability. As PKC has been reported to be activated incessantly in diabetic vessels [161], the reduced NO bioavailability found in the endothelium may be mediated by MAPK/NFkB pathways. Tabit and colleagues [162] reported that PKCβ was upregulated and linked to endothelial dysregulation in diabetic conditions.

In addition, it is also found that inhibition of p38 and JNK brought down the production of superoxide in a manner identical to the of inhibition of NADPH oxidase [160]. While it has been well understood that NADPH oxidase does play a role in activating the stress-induced MAPK p38 and JNK via excessive production of ROS [163,164], a study found that activation of p38 itself could also promote NADPH oxidase activation, thus further worsening oxidative stress in vascular endothelial tissues [165]. Considering that both aPKCs and nPKCs were also found to directly phosphorylate NADPH oxidase, specifically on its p47phox and p60phox subunits [166], this suggests a tight crosstalk between PKCs and MAPKs pathways, especially in promoting endothelial dysfunction via oxidative stress.

Mediating via ERK1/2 cascade, PKC stimulates the entry of arginine into endothelial cells by inducing cationic amino acid transporters 2 (CAT2) arginine transporter. Consequently, this increases arginase expression and activity in the endothelial cells, hence shifting arginine metabolism from NO synthesis to ornithine and urea production, further enhancing eNOS phosphorylation, and thus reducing NO production [167]. In addition, p38 also stimulates inducible NO synthase (iNOS) expression, producing superoxide anions that react with NO into oxidant peroxynitrite, further exacerbating endothelial oxidative stress [168].

Meanwhile, in the kidney, MAPKs were found to be activated in response to high glucose, thereby inducing an inflammatory response in the tubular epithelial cells [169]. While previous studies inferred that angiotensin II could activate MAPKs in the diabetic kidney and renal cells [170,171], other studies showed that the activation of MAPKs themselves may regulate angiotensin-converting enzyme (ACE) gene expression. The promotion of ACE, in turn, activates the renin-angiotensin system (RAS) and further promotes the phosphorylation of ERK1/2, JNK and p38 [172,173,174]. In addition, a previous study suggested that PKC-MAPKs pathways mediated the upregulation of ACE [172]. As MAPKs are known to mediate renal nephropathy by promoting inflammation, macrophage infiltration, and subsequent renal dysfunction, inhibiting these kinases effectively prevented the development of diabetic nephropathy [175]. Pan and colleagues [176] further confirmed that phosphorylation of MAPKs indeed promotes ACE by inducing activator protein-1 (AP-1) activity.

### 5.4. PKC-MAPK Pathway in Atherosclerosis

Persistent hyperglycaemia in diabetic conditions could induce the development of atherosclerosis as well as exaggerate its progression [177]. Atherosclerosis is a widespread chronic inflammatory disorder that affects the arterial wall. It is characterised as a lesion of the intimal layer of the arterial wall and accumulation of plaques which could trigger potentially fatal thrombotic events upon its erosion or rupture [177]. In addition, atherosclerosis is typically associated with impairment in lipid metabolism and hypercholesterolemia accompanied by high low-density lipoprotein (LDL) in the blood, heightening the risk for cardiovascular diseases [178,179]. Pathogenesis-wise, the development and progression of this condition involve the interplay of various cell types, including endothelial cells, vascular smooth muscle cells (VSMCs), macrophages, and other immune cells [180]. Inflammatory mediators and leukocyte adhesion molecules such as intercellular adhesion molecule-1 (ICAM-1) and vascular cell adhesion molecule-1 (VCAM-1) attract migration and infiltration of monocytes into the subendothelial space. These monocyte-derived macrophages are involved in forming foam cells along with the uptake of LDL or oxLDL. The foam cells and oxLDL accumulate in the intima, generating oxidative stress and releasing platelet-derived growth factors that trigger the switch of VSMC phenotype. The switching of VSMC results in necrotic core formation and inflammation and produces extracellular matrix in the fibrous cap [181].

In the vascular wall, the VSCMs typically display a contractile phenotype which is crucial for maintaining vascular tone. However, VSMCs can differentiate upon stimulation into a synthetic phenotype, and this phenotype switching process is a critical mechanism in arterial remodelling [182,183]. The difference between the phenotype is that in the contractile phenotype, the VSMCs contain high levels of contractile genes such as α-smooth muscle actin (αSMA), smooth muscle 22α (SM22α), and heavy-caldesmon (h-Cad) as well as having low proliferation rates, migration, and extracellular matrix production. On the other hand, synthetic VSMCs contain low contractile gene expression and higher rates of proliferation, migration, and extracellular matrix production [184]. In this regard, PKC was suggested to mediate apoptosis of proliferated synthetic VSMCs and continuous uptake of oxLDL [185]. Even though apoptosis of the synthetic VSMCs may reduce intimal hyperplasia that promotes the formation of atherosclerotic plaque, it also increases the risk of plaque instability and rupture [33].

As in many other cells, PKC activation is associated with phosphorylation of ERK1/2 in the vascular smooth muscle cells [186,187]. Hu et al. [188] reported that smooth muscle cells in lesions expressing activated ERK1/2. Abundant ERK1/2 proteins were observed in the atherosclerotic lesions. ERK1/2 protein in intima and media was also significantly higher than in control. In in vivo study, ERK1/2 activation was seen sharply elevated with lesions, suggesting that hyperexpression of these kinases may play a central role in regulating cell proliferation in the pathogenesis of atherosclerosis.

Moreover, PKC is also involved in the signalling pathways that mediate the different roles of macrophages during atherogenesis. Upregulation of inflammatory cytokines by resistin in the macrophages is also mediated by PKCε [189]. In the vascular endothelial and smooth muscle cells, JNK phosphorylation also triggers the release of pro-inflammatory cytokines and promotion of plaque formation downstream of stressors and PKC activation [190,191,192]. On top of that, PKCδ via ERK signalling pathways was found to increase inflammatory cytokines expression, oxLDL uptake and foam cell formation in macrophages [193]. PKCδ can be found highly expressed in human atherosclerotic arteries and CD68-positive macrophages; it shows that PKCδ indeed plays a significant role in the formation of foam cells.

## 6. PKC-MAPK as Therapeutic Target for Cardiovascular Management in Diabetic Patients

Preventing the development and progression of cardiovascular complications is the primary goal of cardiovascular management in diabetic patients. As the PKC-MAPK pathway has shown to play a significant role in the development of diabetes-induced cardiovascular illnesses, it may serve as a potential therapeutic target to manage this issue in diabetic patients. Applying PKC and MAPK modulation in clinical trials aimed to retard cardiovascular complications has been attempted with mixed results in recent years. Promising results have been observed in using the PKCβ inhibitor ruboxistaurin, whereby its usage was found to protect against diabetic nephropathy progression in type 2 diabetes patients as reflected by reduced albuminuria and controlled eGFR [194]. Similar results were replicated by Gilbert et al. [195] and Casellini et al. [196], showing reduced urinary levels of TGF-β, suggesting truncation of renal fibrosis and diabetic nephropathy; these findings indeed demonstrated the promising potential of PKC inhibitors to be used to protect against diabetes-derived microvascular complications. Regarding cardiac protection, albeit lack of clinical trials, cardiac function improvements were observed on the use of ruboxistaurin on the porcine model of heart failure [197] and require clinical testing to support its use in humans. 

However, using ERK1/2 inhibitors in clinical trials seems to be discouraging for managing cardiac function. A few studies showed that instead of exerting protective effects, inhibiting ERK1/2 instead increases the risk of hypertension and cardiac function disturbance [198,199]. This is probably due to the fact that ERK1/2 also plays a role in regulating cell proliferation, which is a crucial function, especially in post-traumatic events such as myocardial infarction and ischemic reperfusion injury. Nevertheless, evidence of its effect in the diabetic setting has been scarce. As promising results were revealed in a pre-clinical study in a diabetic model [200], this warrants a clinical investigation to explore this possible protective effect of modulating ERK1/2 activation with extra precautions given to its adverse effects too.

In contrast, modulating stress MAPKs p38 and JNK with their specific inhibitors seems to be more effective in clinical findings. Treatment with p38 inhibitors BMS-582949 [201] and losmapimod [202] were both well-tolerated in patients, showing no adverse effects on atherosclerotic and myocardial infarcted patients with reduced hypertrophy markers observed. In fact, losmapimod treatment was found to improve nitric oxide-mediated vasodilatation, showing that p38 inhibition by this drug was able to improve nitric oxide activity in line with pre-clinical findings [203]. Losmapimod was also found to reduce vascular inflammation in atherosclerotic patients [204]. However, the use of JNK-specific inhibitor is lacking and not well-explored. In fact, only a few JNK inhibitors have been tested in clinical trials for various indications, yet none for the treatment of cardiovascular pathology. Preclinical studies on the use of JNK-specific inhibitors such as AS601245 and SP600125 have been found to exert cardiovascular protective effects by attenuating cardiac inflammation, endothelial dysfunction, as well as protecting against diabetic nephropathy progression [205,206,207].

Even so, the majority of the research on the use of PKC-MAPK inhibitors has been unable to identify the isoenzymes of interest, as well as the specific downstream signalling events that lead to cardiovascular complications. Despite a great deal of evidence from preclinical studies that support the modulation of this pathway independently or in broad, the experimental model used does not mimic well the disease progression in humans, thus affecting the reproducibility of the positive results in humans. Nevertheless, given the importance of PKC-MAPK signalling pathways in both normal physiology and disease pathogenesis, these kinase families remain an appealing target for drug development.

## 7. Conclusions

Our review suggests that PKC-MAPK pathways play an extensive role in the pathophysiology of diabetes-induced cardiovascular complications and hence may serve as an excellent therapeutic target for cardiovascular management. Identifying the critical PKC isoforms involved in the activation of MAPKs that exert pathogenesis, and the determination of potential synergistic effects of the combinations of isoform-specific inhibitors in the treatment of diabetes-associated cardiovascular complications, are two potential challenges in developing therapeutic agents targeting the PKC-MAPK pathway. Although further confirmatory in vitro, in vivo, and human studies are required, we anticipate that the promising role of the PKC-MAPK pathway can potentiate the treatment of diabetes-associated cardiovascular complications in human.

## Figures and Tables

**Figure 1 ijms-23-08582-f001:**
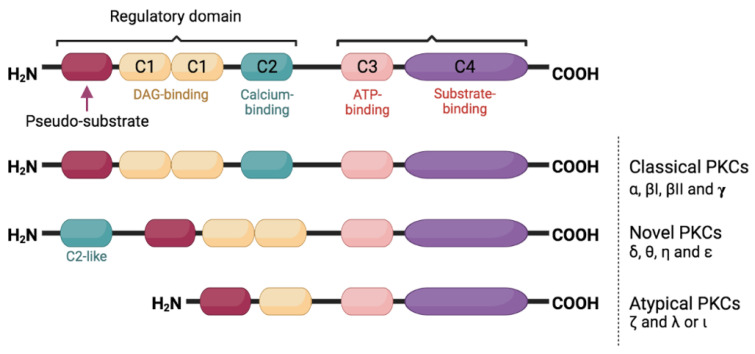
Classification and structural characteristics of PKC isoforms. Although the catalytic domain of PKC is conserved, the three subgroups have different regulatory domains. The classical PKC isoforms (cPKC) share all typical regulatory features: the autoinhibitory pseudosubstrate motif, two DAG-binding C1 domains and the calcium-binding C2 domain. Novel PKC isoforms (nPKC) lack a calcium-binding motif but contain an extended N-terminal domain that can receive regulatory signals, and they are still being regulated by DAG. On the other hand, the catalytic activity of atypical PKC isoforms (aPKC) is independent of DAG and calcium.

**Figure 2 ijms-23-08582-f002:**
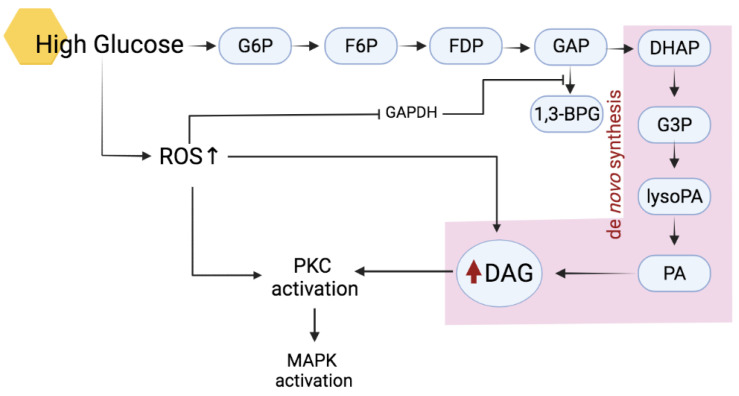
Mechanism of DAG–PKC activation. Hyperglycaemia induces mainly the de novo pathway of DAG synthesis from glucose to glycerol 3-phosphate with subsequent PKC activation. ROS, reactive oxygen species; DAG, diacylglycerol; DHAP, dihydroxyacetone phosphate; FDP, fructose 1,6-diphosphate; F6P, fructose 6-phosphate; GAP, glyceraldehyde 3-phosphate; G3P, glycerol 3-phosphate; G6P, glucose 6-phosphate; lysoPA, lysophosphatidic acid; PA, phosphatidic acid; PKC, protein kinase C; MAPK, mitogen-activated protein kinase. Arrows represent the subsequent event/product in the pathway; red arrow represents increased production.

**Figure 3 ijms-23-08582-f003:**
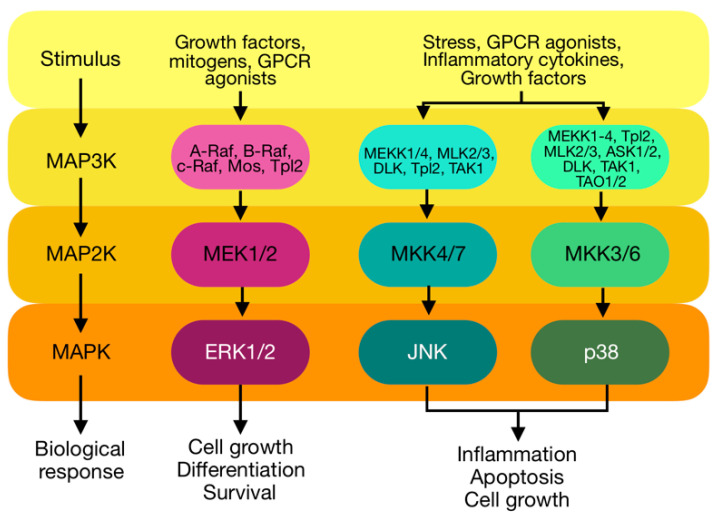
MAPKs cascades: MAPK signalling cascades are organized hierarchically into three-tiered modules. MAPKs are phosphorylated and activated by MAPK-kinases (MAP2Ks), which in turn are phosphorylated and activated by MAPKK-kinases (MAP3Ks). The MAP3Ks are in turn activated by interaction with the family of small GTPases and/or other protein kinases, connecting the MAPK module to cell surface receptors or external stimuli.

**Figure 4 ijms-23-08582-f004:**
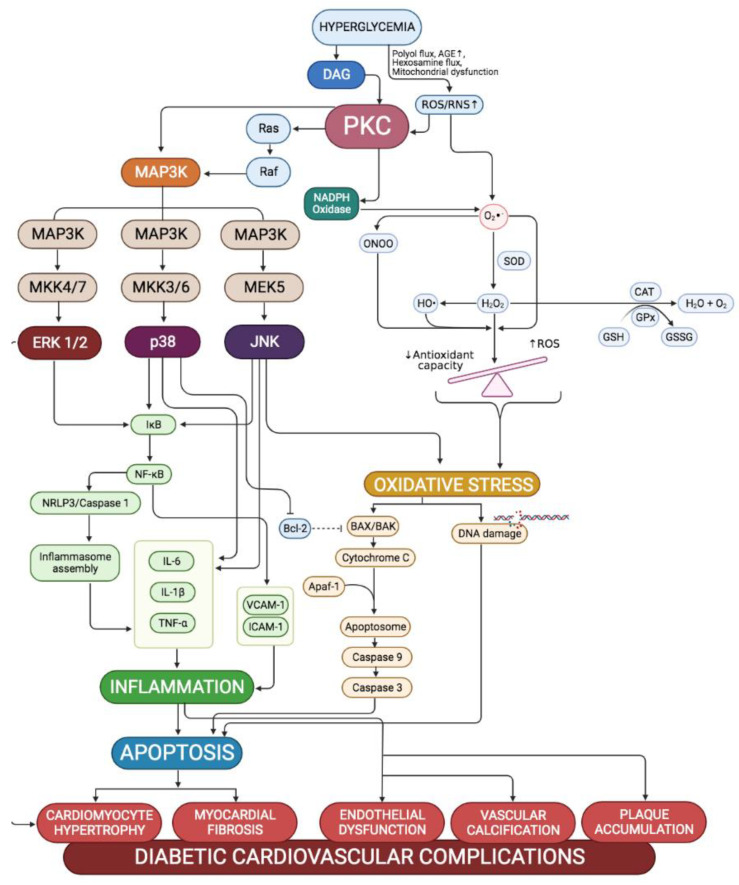
Proposed mechanisms involved in the pathogenesis of diabetic cardiovascular complications via the PKC-MAPK pathway. Persistent hyperglycemia induces de novo production of diacylglycerol (DAG), which along with reactive oxygen (ROS), would induce the activation of PKC. Hyperglycemia also generates ROS and nitrogen species (RNS) in excess via NADPH oxidase, generating oxidative stress in the cardiac and vascular tissues. By activating Raf dependently and independently of Ras activation, PKC phosphorylates MAP3Ks of the MAPK subfamilies cascades. Following a stimulation, activation of MAPK requires a three-tiered kinase cascade; in which a MAPK kinase kinase (MAPKKK, MEKK, MAP3K or MKKK) activates a MAPK kinase (MAPKK, MEK, MAP2K or MKK), which in turn activates the targeted MAPKs (ERK1/2, JNK and p38) through serial phosphorylation. Uncontrolled activation of MAPKs induces inflammation and cell apoptosis, leading to pathologic cardiac remodelling and vascular dysfunction. Arrows represent the subsequent event/product in the pathway; tappered line represents inhibition action; dotted tappered line represents halted inhibition action.

## Data Availability

No new data were created or analyzed in this study. Data sharing is not applicable to this article.

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
