# Peer review of "The Role of PKC-MAPK Signalling Pathways in the Development of Hyperglycemia-Induced Cardiovascular Complications"

_ijms, 2022, doi:10.3390/ijms23158582_

Round 1

Reviewer 1 Report

I found the review of interest and well written but I have noted many improvements to make in the use of English which spoil the reading even though they are minor because they are numerous. In a few places I did not exactly understand what was meant. Reference 183 seems to be missing but otherwise I have no major faults but please check against my attached pdf with comments and corrections suggested.. 

Author Response

We appreciate the time and effort you and the reviewers dedicated to providing feedback on our manuscript. We are grateful for the insightful comments on and valuable improvement to our paper. We have incorporated most of the suggestions made by the reviewers. Those changes are highlighted within the manuscript. Please see in blue below for a point-by-point response to the reviewers’ comments and concerns. All page numbers refer to the revised manuscript file with tracked changes.

Comments and Suggestions for Authors

Reviewer #1

I found the review of interest and well written but I have noted many improvements to make in the use of English which spoil the reading even though they are minor because they are numerous. In a few places I did not exactly understand what was meant. Reference 183 seems to be missing but otherwise I have no major faults but please check against my attached pdf with comments and corrections suggested.

Author response: Thank you very much for your comments and insights. As suggested, we went through the manuscript for grammatical errors, and fully English proofread the manuscript.

We found that there has been an issue with the referencing numbering and list, which causes reference 183 to be missing. We have rectified this issue.

Thank you very much for your time. We genuinely appreciate it.

Best regards,

Fatin Farhana Jubaidi

Reviewer 2 Report

The review is focused on PKC-MAPK signaling pathways and their role in development of hyperglycemia-induced cardiovascular complications. From this point is the review focused on an actual and important topic.

I have several questions and comments:

1. In several cases are discrepancies in references. Here are only some examples: 

“Cheng et al. [68] reported that PKC-α and PKC-ɛ act as Raf-1 activators that lead to a prolonged effect on the ERK1/2 signaling pathway.” The reference (citation) is not correct. Number 68 has a paper of Zhang et al. 2005.  

“Huang et al. [152] demonstrated in their study that endothelial dysfunction in diabetic mice were found to be mediated by both p38 and JNK activation with significant impairment in endothelial nitric oxide synthase (eNOS).” Citation 152 is Tabit et al. (2013), Huang et al. is reference 151.

“Hu et al. [178] reported that smooth muscle cells in lesions expressing activated ERK1/2.” This work of Hu et al. is not in list of references.

All references should be controlled and corrected.

2. In lines 338-340 you wrote: “Activated PKC would lead to generation of reactive oxygen species (ROS) via activation of nicotinamide adenine dinucleotide phosphate (NADPH) oxidase [92].”

In this case is cited only review article focused on similar topic as present review and not original experimental work showing the relation between PKC and (NADPH) oxidase.

3. In lines 375-378 you wrote: “Persistent hyperglycemia was found to induce the expression of these cytokines in the heart by activating the MAPK pathways (JNK and p38-MAPK), causing cardiac damage, which further aggravates the infiltration and accumulation of leukocytes onto the affected site [103].”

The cited work is not dealing with activation of MAPK pathways in consequence of  hyperglycemia and cytokines induction.

4. In lines 381-382 you wrote: “Inhibition of Nrf-2 by ERK 1/2 exaggerates oxidative stress, which in turn activates NF-κB [105].”

The used reference is not correct. The paper Baker et al. (2011) did not bring information about inhibition of Nrf-2 by ERK 1/2 and the relation between Nrf-2, ERK, and NF-κB.

5. In lines 451-453 you wrote: “Overexpression of either wild-type or a constitutively active deletion mutant of PKCβ in a mouse heart was reported to induce cardiomyopathy [124].”

Is the used reference correct? Study of Braz et al. (2004) deals with PKC-alpha.

6. In lines 468-469 you wrote: “Initial studies in cultured neonatal cardiomyocytes indicated that overactivation of JNK by MKK7, an upstream MAP2K, leads to a hypertrophic phenotype [127].”

Is the used reference correct? Study of Wang et al. (1998) deals with p38-MAPK and is not focused on JNKs.

7. Chapter 5. of review is focused on involvement of PKC-MAPK pathway in diabetic heart complications.

However, the role of PKC/MAPK pathway in cardiac hypertrophy (chapter 5.1), in cardiac fibrosis (chapter 5.2), and also in atherosclerosis (chapter 5.4) is not described in direct relation to diabetes and hyperglycemia (topic of your review article).

8. In lines 24-27 (Abstract) you wrote: “In this review, we discuss the recent discovery on the role of PKC/MAPK pathways and the mechanisms involved in the development and progression of diabetic cardiovascular complications and its potential as therapeutic targets for the cardiovascular management in diabetic patients.”

The potential PKC/MAPK pathways as therapeutic targets for the cardiovascular management in diabetic patients is discussed very weakly.  

Author Response

We appreciate the time and effort that you and the reviewers dedicated for providing feedback on our manuscript and are grateful for the insightful comments on and valuable improvement to our paper. We have incorporated most of the suggestions made by the reviewers. Those changes are highlighted within the manuscript. Please see below, in blue, for a point-by-point response to the reviewers’ comments and concerns. All page numbers refer to the revised manuscript file with tracked changes.

Comments and Suggestions for Authors

Reviewer #2

The review is focused on PKC-MAPK signaling pathways and their role in development of hyperglycemia-induced cardiovascular complications. From this point is the review focused on an actual and important topic.

I have several questions and comments:

  1. In several cases are discrepancies in references. Here are only some examples:

“Cheng et al. [68] reported that PKC-α and PKC-ɛ act as Raf-1 activators that lead to a prolonged effect on the ERK1/2 signaling pathway.” The reference (citation) is not correct.

Number 68 has a paper of Zhang et al. 2005.

“Huang et al. [152] demonstrated in their study that endothelial dysfunction in diabetic mice were found to be mediated by both p38 and JNK activation with significant impairment in endothelial nitric oxide synthase (eNOS).” Citation 152 is Tabit et al. (2013), Huang et al. is reference 151.

“Hu et al. [178] reported that smooth muscle cells in lesions expressing activated ERK1/2.” This work of Hu et al. is not in list of references.

All references should be controlled and corrected.

Author response: Thank you very much for pointing out this mistake. We found that there has been an issue with the referencing numbering and list, which causes the reference numbering to be missing and misplaced. We have rectified this issue.

2. In lines 338-340 you wrote: “Activated PKC would lead to generation of reactive oxygen species (ROS) via activation of nicotinamide adenine dinucleotide phosphate (NADPH) oxidase [92].”

In this case is cited only review article focused on similar topic as present review and not original experimental work showing the relation between PKC and (NADPH) oxidase.

Author response: Thank you very much for pointing out this mistake. We found that there has been an issue with the referencing numbering and list, which causes the reference numbering to be missing and misplaced. We have rectified this issue.

3. In lines 375-378 you wrote: “Persistent hyperglycemia was found to induce the expression of these cytokines in the heart by activating the MAPK pathways (JNK and p38-MAPK), causing cardiac damage, which further aggravates the infiltration and accumulation of leukocytes onto the affected site [103].”

The cited work is not dealing with activation of MAPK pathways in consequence of hyperglycemia and cytokines induction.

Author response: Thank you very much for pointing out this mistake. We found that there has been an issue with the referencing numbering and list, which causes the reference numbering to be missing and misplaced. We have rectified this issue.

4. In lines 381-382 you wrote: “Inhibition of Nrf-2 by ERK 1/2 exaggerates oxidative stress, which in turn activates NF-κB [105].”

The used reference is not correct. The paper Baker et al. (2011) did not bring information about inhibition of Nrf-2 by ERK 1/2 and the relation between Nrf-2, ERK, and NF-κB.

5. In lines 451-453 you wrote: “Overexpression of either wild- type or a constitutively active deletion mutant of PKCβ in a mouse heart was reported to induce cardiomyopathy [124].”

Is the used reference correct? Study of Braz et al. (2004) deals with PKC-alpha.

Author response: Thank you very much for pointing out this mistake. We found that there has been an issue with the referencing numbering and list, which causes the reference numbering to be missing and misplaced. We have rectified this issue accordingly.

6. In lines 468-469 you wrote: “Initial studies in cultured neonatal cardiomyocytes indicated that overactivation of JNK by MKK7, an upstream MAP2K, leads to a hypertrophic phenotype [127].”

Is the used reference correct? Study of Wang et al. (1998) deals with p38-MAPK and is not focused on JNKs.

Author response: Thank you very much for pointing out this mistake. We found that there has been an issue with the referencing numbering and list, which causes the reference numbering to be missing and misplaced. We have rectified this issue.

7. Chapter 5. of review is focused on involvement of PKC-MAPK pathway in diabetic heart complications.

However, the role of PKC/MAPK pathway in cardiac hypertrophy (chapter 5.1), in cardiac fibrosis (chapter 5.2), and also in atherosclerosis (chapter 5.4) is not described in direct relation to diabetes and hyperglycemia (topic of your review article).

Author response: Thank you very much for your comment and insight. We have added the information to support the roles of PKC/MAPK pathways in these subheadings accordingly.

8. In lines 24-27 (Abstract) you wrote: “In this review, we discuss the recent discovery on the role of PKC/MAPK pathways and the mechanisms involved in the development and progression of diabetic cardiovascular complications and its potential as therapeutic targets for the cardiovascular management in diabetic patients.”

The potential PKC/MAPK pathways as therapeutic targets for the cardiovascular management in diabetic patients is discussed very weakly.

Author response: Thank you very much for your insightful comment. We have added a new subheading (6. PKC/MAPK as therapeutic target for cardiovascular management in diabetic patients; page 14) to address this issue.

Thank you very much for your time. We genuinely appreciate it.

Best regards,

Fatin Farhana Jubaidi

Round 2

Reviewer 2 Report

Authors included corresponding changes to the revised manuscript.